# Clinical factors associated with progression to dementia in people with late-life depression: a cohort study of patients in secondary care

Georgia Peakman [ID],[1] Nishshanka Karunatilake,[2] Mathieu Seynaeve,[1,2] Gayan Perera [ID],[1] Dag Aarsland,[1,3] Robert Stewart,[1,2] Christoph Mueller[1,2]

[1]King's College London, Institute of Psychiatry, Psychology and Neuroscience, London, UK
[2]South London and Maudsley NHS Foundation Trust, London, UK
[3]Centre for Age-Related Medicine, Stavanger University Hospital, Stavanger, Norway

**Correspondence to**
Georgia Peakman;
georgia.peakman.18@ucl.ac.uk

## ABSTRACT

**Objectives** Depression can be a prodromal feature or a risk factor for dementia. We aimed to investigate which clinical factors in patients with late-life depression are associated with a higher risk of developing dementia and a more rapid conversion.

**Design** Retrospective cohort study.

**Setting** South London and Maudsley NHS Foundation Trust (SLaM) secondary mental healthcare services.

**Participants** The SLaM Clinical Record Interactive Search was used to retrieve anonymised data on 3659 patients aged 65 years or older who had received a diagnosis of depression in mental health services and had been followed up for at least 3 months.

**Outcome measures** Predictors of development of incident dementia were investigated, including demographic factors, health status rated on the Health of the National Outcome scale for older people (HoNOS65+), depression recurrence and treatments including psychotropic drugs and cognitive behavioural therapy (CBT).

**Results** In total, 806 (22.0%) patients developed dementia over a mean follow-up time of 2.7 years. Significant predictors of receiving a dementia diagnosis in fully adjusted models and after accounting for multiple comparisons were older age (adjusted HR=1.04, 95% CI 1.03 to 1.06 per year difference from sample mean) and the HoNOS65+ subscale measuring cognitive problems (HR=4.72, 95% CI 3.67 to 6.06 for scores in the problematic range). Recurrent depressive disorder or past depression (HR=0.65, 95% CI 0.55 to 0.77) and the receipt of CBT (HR=0.73 95% CI 0.61 to 0.87) were associated with a lower dementia risk. Over time, hazards related to age increased and hazards related to cognitive problems decreased.

**Conclusions** In older adults with depression, a higher risk of being subsequently diagnosed with dementia was predicted by higher age, new onset depression, severity of cognitive symptoms and not receiving CBT. Further exploration is needed to determine whether the latter risk factors are responsive to interventions.

## INTRODUCTION

A link between depression and dementia is well established, and growing evidence

## Strengths and limitations of this study

► This study used a large naturalistic sample of older patients receiving routine mental healthcare, with access to anonymised clinical data.
► The sample was limited to patients who had been diagnosed with depression, followed up and, if applicable, diagnosed with dementia within the same secondary provider; it is not readily generalisable to cases of depression treated in other settings.
► Time between depression diagnosis and dementia diagnosis is additionally influenced by potential barriers to seeking a dementia diagnosis such as service, provider, patient and caregiver factors, which were not captured by this analysis.
► The influences of other comorbid long-term conditions and resultant polypharmacy were not fully considered, and causal inferences cannot be readily made from observational data.

indicates that the timing of depression is important in defining the nature of the relationship.[1] Late-life depression is projected to increase substantially by 2050[2] and has consistently been associated with up to a twofold increased risk for dementia,[3–5] and it is now recognised as a key potentially modifiable risk factor for the condition.[6] However, causality is yet to be established; it remains unclear whether, and in which cases, late-life depression is an independent risk factor for dementia or a prodromal symptom.

Identification of the clinical characteristics of older adults with depression that are indicative of incipient dementia would facilitate accurate classification of individuals at high risk for early progression and subsequently, timelier treatment. The pattern and severity of depressive symptoms in late life has emerged as one important factor in determining the likelihood of progression to dementia. In two large, longitudinal cohort studies of older adults, those subjects characterised

by trajectories of high and steadily increasing depressive symptoms over several years were significantly more likely to develop dementia.[7 8] Having an episode of high depressive symptoms that remitted was not associated with greater dementia risk.[8] Similarly, recent data from the Whitehall II cohort study characterised those with late-life depression who later developed dementia as having a distinguishable trajectory of increasing depressive symptoms, emerging 11 years before dementia diagnosis.[9] Depressive symptoms during mid-life, even when recurrent, did not increase dementia risk in this cohort. These findings, in addition to other reports of graded associations with depression severity and raised dementia risk being contingent on the two being in closer temporal proximity,[10 11] lend support to the hypotheses that depressive symptoms during later life can be an early manifestation of dementia,[12] or that the two conditions share common aetiological factors[13 14]; potentially both are consequences of an underlying neurodegenerative process.[15]

While it is well established that greater severity of depressive symptoms is closely associated with accelerated development of dementia, other associations relevant to this relationship are less clear. Using naturalistic data from a cohort of secondary mental health service users, the present study examined associations between patient characteristics and interventions within late-life depression and the risk of progression to a diagnosis of dementia.

## METHODS
### Source of data
Data for this study were obtained via the South London and Maudsley NHS Foundation Trust (SLaM), one of Europe's largest mental healthcare providers, serving 1.3 million residents in four south London boroughs (Croydon, Lambeth, Lewisham and Southwark), as well as delivering specialist national services. SLaM adopted fully electronic records across all of its services from 2006, having imported legacy systems which include a records system run for several years previously in dementia care. The Clinical Record Interactive Search (CRIS) application was developed in 2008 to provide access to deidentified structured and open-text data extracted from this electronic health records system, containing records on over 400 000 SLaM mental health service users.[16 17] CRIS has been enhanced by a range of natural language processing ('text-mining') applications[18] which extract structured data from text fields, including applications to ascertain pharmacotherapy.[17] CRIS has research ethical approval as an anonymised database for secondary analysis (Oxford REC C 08/H0606/71+5).

### Sample
Using CRIS, people aged 65 or older who received a diagnosis of depression from SLaM services within the 11-year period between 1 January 2006 and 30 March 2017 were

identified according to the International Statistical Classification of Diseases and Related Health Problems 10th Revision (ICD-10)[19] criteria described below. Date of first depression diagnosis after the age of 65 years served as the index date. Patients were followed up until the date of their first dementia diagnosis (incident dementia) given in the health record; or their last face-to-face contact with a professional from secondary mental health services (clinical event or appointment); or death; or a censoring point on 30 June 2017. The censored times therefore correspond to all events bar the occurrence of the patient's first diagnosis. Patients were excluded if they had a diagnosis of dementia recorded before or within 3 months of the index date, or if they had less than 3 months of contact with SLaM services after their depression diagnosis.

### Measurements
Cases of depression and dementia according to ICD-10 codes were ascertained, both from diagnoses recorded in structured fields and in free text within clinical correspondence, through a natural language processing algorithm.[18]

#### Late-life depression
Initial depression diagnoses (the index contact) were determined by the following diagnostic codes from ICD-10[19] within a primary or secondary diagnosis: F32 (depressive episode), F32.2 (severe depressive episode without psychotic symptoms), F32.3 (severe depressive episode with psychotic symptoms), F33 (recurrent depressive disorder), F33.2 (recurrent depressive disorder, current episode severe without psychotic symptoms), F33.3 (recurrent depressive disorder, current episode severe with psychotic symptoms). Data were also obtained for any depression diagnoses (F32, F33) recorded before the index date.

#### Dementia
Dementia diagnoses were determined by the following diagnostic codes from ICD-10[19] within a primary or secondary diagnosis: F00 (dementia in Alzheimer's disease (AD)), F01 (vascular dementia), F02 (dementia in other diseases classified elsewhere), F03 (unspecified dementia).

#### Covariates
Data were also extracted on age (at depression diagnosis), gender, ethnicity, marital or cohabiting status, local authority, a neighbourhood-level deprivation score (Index of Multiple Deprivation, 2010),[20] psychotropic medication (classified into antipsychotic, antidepressant and selective serotonin reuptake inhibitor (SSRI) types) prescribed in the period 2 years prior to depression diagnosis and following diagnosis, and the receipt of cognitive behavioural therapy (CBT) after depression diagnosis. Age was centred in the regression models, meaning that incremental effects related to a 1-year difference from the mean. The Health of the Nation Outcome Scales for

older people[21] (HoNOS65+) is routinely administered in SLaM services, and the subscale scores available closest to the depression diagnosis extracted for this analysis were agitated behaviour, non-accidental self-injury, drug/alcohol problems, hallucinations or delusions, depressed mood, physical illness or disability, cognitive problems, activities of daily living (ADLs), living conditions, occupational and recreational activities, social relationships. HoNOS65+ items are rated from 0 (least severe status) to 4 (most severe status). For the purpose of this analysis, scores of 0–1 were considered to indicate that the patient was not experiencing problems in the domain, and scores of 2–4 were considered to indicate experiencing severe problems as rated by the clinician.

## Statistical analysis

Baseline characteristics of the cohort grouped according to dementia development (those who were diagnosed with dementia within secondary mental health services and those who did not have a dementia diagnosis at their last contact with services) were compared using $\chi^2$ tests for categorical variables, and independent samples t-tests or analysis of variance for continuous variables.

Cox proportional hazards regression models were used to analyse associations between the predictor variables at depression diagnosis and hazard of receiving a diagnosis of dementia. Multivariate models were first adjusted for demographic factors; age, gender and deprivation (Model 1), then for the HoNOS65+ variables of cognitive problems and depressed mood (Model 2), and finally all variables were adjusted for in Model 3 to determine independent associations. All analyses were performed using Stata V.13.0.[22] As 19% of the cohort had missing data on at least one covariate, we used the *mi* package in STATA to create 19 imputed datasets, generated by replacing missing values with simulated values assembled from potential covariates and outcome values.[23] Rubin's rules[24] were applied to combine coefficients in final analyses. We assessed the proportional-hazard assumptions for each variable on the basis of Schoenfeld residuals in the complete-case sample and included variables as time-varying if they were violated (both in the complete-case and imputed sample). We report results at both a significance level of p<0.05 and a Bonferroni-corrected p<0.002 to account for multiple comparisons (22 variables examined). In a second step, we only considered those patients in the cohort who had a diagnosis of dementia recorded to determine how the duration from depression to dementia diagnosis varied according to the predictors that had yielded significance in the above Cox regression models.

## Patient and public involvement

A patient-led committee provides operational oversight of the CRIS resource and ensures that the proposed study has both scientific value and benefit for patients and carers.[17 25] The authors had applied to the oversight committee with an outline of the proposed analysis,

including data linkages, and the project was approved on 31 May 2017. There was no further patient and public involvement in this register-based study.

## RESULTS

### Sample size and characteristics

During the 11-year study period, a total of 6670 patients diagnosed with depression at age 65 or above were identified (figure 1). Of these, 887 patients were excluded because they already had a comorbid diagnosis of dementia at the time or within 3 months of the depression diagnosis. A further 2124 patients were excluded because they were followed up for less than 3 months after their depression diagnosis. The final cohort consisted of 3659 patients with late-life depression, of which 806 (22.0%) received a diagnosis of dementia from mental health services during the follow-up period. Mean follow-up time until dementia diagnosis, last clinical face-to-face contact, census or death was 2.7 years (SD±2.6 years); and this did not differ significantly between those who were or were not diagnosed with dementia (see table 1).

Baseline characteristics of the late-life depression cohort as a whole and according to dementia development are presented in table 1. Patients who were subsequently diagnosed with dementia were significantly older and more likely to be female, have cognitive problems and impairment of their ADLs. They were less likely to have recurrent depression, non-accidental self-injury recorded or problems related to their depressed mood or social relationships. In terms of pharmacotherapy, those who were diagnosed with dementia were more likely to be prescribed any antidepressant, a SSRI or an antipsychotic after depression diagnosis, but were less likely to have received CBT.

### Predictors of receiving a dementia diagnosis in secondary care services

Table 2 presents the findings from a series of multivariate Cox proportional hazards models examining associations between the predictors in late-life depression and risk of dementia diagnosis. Analyses were first adjusted for the demographic factors of age, gender and deprivation (Model 1). In this model, when considering p<0.002 as significant to account for multiple comparisons, older age (as per year differing from the sample mean of 76.0 years) and non-white ethnicity, as well as the HoNOS65+ subscale scores recording problems with physical illness, cognition and ADLs were each significantly associated with a higher risk of developing dementia, while recurrent depression and treatment with CBT were associated with lower dementia risk. Both the protective effect of having recurrent depression and the hazardous effect of cognitive problems lessened over time. In Model 2 and the fully adjusted Model 3 (presented in figure 2), at a significance level of p<0.002, older age and cognitive problems remained associated with progression to dementia, and recurrent depression and treatment with CBT both

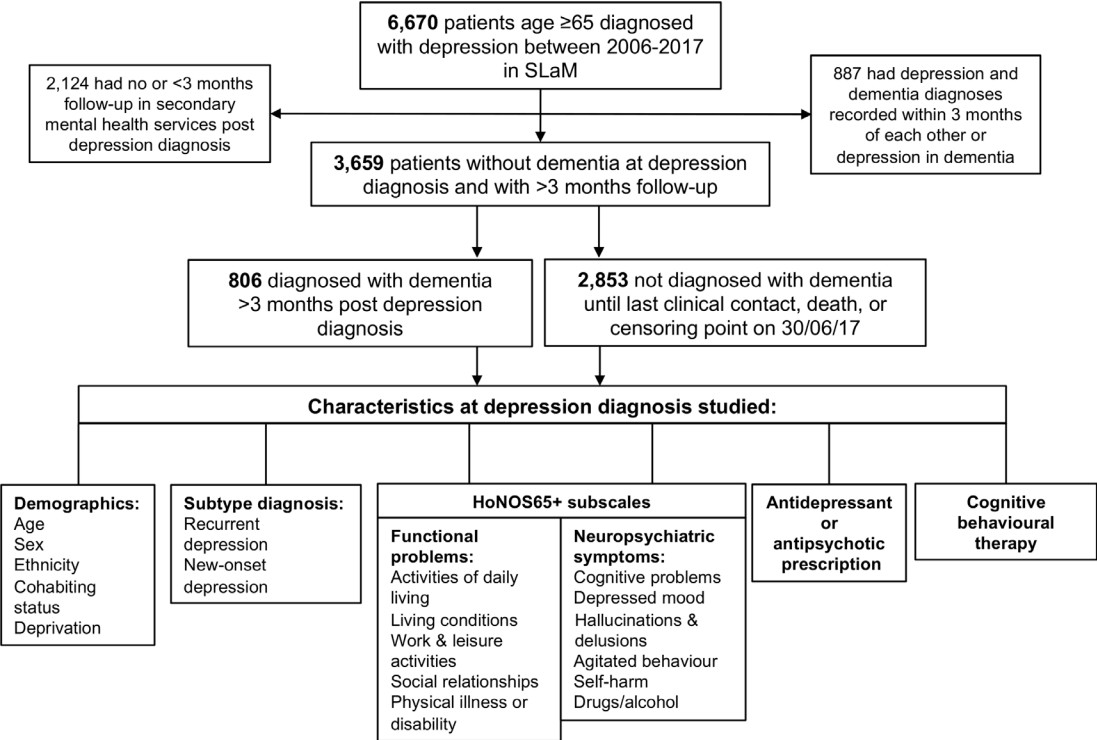

**Figure 1** Flowchart showing cohort generation and outcomes. SLaM, South London and Maudsley NHS Foundation Trust.

remained associated with lower dementia risk. In these two models, the hazard associated with age increased and the hazard related to cognitive problems decreased with time. When applying a significance level of p<0.05 (not corrected for multiple comparisons), presenting with non-accidental self-injury or hallucinations and/or delusions at depression diagnosis were associated with a lower risk of dementia, while those prescribed any antidepressant had a 33% increased dementia risk. The hazard associated with hallucinations and/or delusions increased with time.

Alternatively adding age as a squared term to the analyses did not alter the predictors yielding significance in the fully adjusted model (Model 3), with the exception of non-accidental self-injury (p=0.076) and antidepressant prescription after index date (p=0.056) which both attenuated to a trend. Similar results were found in the complete case sample which included 2952 patients, of whom 717 developed dementia (see online supplementary table 1).

### Characteristics of 806 patients who received a dementia diagnosis

In those who received a dementia diagnosis, the average time from depression diagnosis to dementia diagnosis was 2.8 years (SD±2.4). Of this group, the first recorded dementia subtype was AD for 353 (43.8%), vascular dementia for 199 (24.7%), dementia in in other diseases for 51 (6.3%) and unspecified dementia for 203 (25.2%) patients.

Figure 3 presents variations in the mean time between depression and dementia diagnosis, according to the

factors that were significantly associated with dementia risk in the fully adjusted model (Model 3) after accounting for multiple comparisons. An age above 78 years at the point of depression diagnosis (the mean age of the patients who were diagnosed with dementia during follow-up) translated to receiving a dementia diagnosis 9 months earlier than average. Having a diagnosis of recurrent depressive disorder or a previous diagnosis of depression were associated with a dementia diagnosis almost 14 months later than average, and treatment with CBT to 11 months later. Presenting with cognitive problems recorded on the HoNOS65+ instrument translated to a dementia diagnosis almost 17 months sooner than average. Table 3 shows time to dementia diagnosis according to severity of cognitive problems rated by clinicians on the HoNOS65+ subscale. Those who were scored to have no or minor cognitive problems at the time of their depression diagnosis progressed to a diagnosis of dementia in 3–4 years, and those with moderate to severe problems in less than 2 years.

### DISCUSSION

This study showed in a cohort of over 3500 older adults seen in secondary mental health services for depression that presenting at an older age, with new-onset depression and more severe cognitive problems, and not being treated with CBT, were each predictive of being subsequently diagnosed with dementia.

There is ongoing debate as to whether recurrent depression[26 27] or history of depression[28] carry an increased risk

**Table 1** Baseline characteristics of the late-life depression cohort according to whether a dementia diagnosis was established

| Baseline characteristics | Whole cohort | Dementia diagnosis >3 months postdepression diagnosis | No dementia diagnosis during follow-up | P value* |
|---|---|---|---|---|
| No. of patients | 3659 | 806 | 2853 | |
| Mean age (SD)† | 76.0 (7.7) | 78.3 (7.2) | 75.3 (7.7) | <0.001 |
| Mean time in years from depression diagnosis to census (dementia/death/last clinical contact) (SD) | 2.7 (2.6) | 2.8 (2.4) | 2.7 (2.6) | 0.365 |
| Female (%) | 64.1 | 68.7 | 62.7 | 0.002 |
| Non-white ethnicity (%) | 17.6 | 19.4 | 17.1 | 0.121 |
| Married or cohabiting status (%)† | 32.1 | 29.6 | 32.8 | 0.091 |
| Mean index of deprivation (SD)† | 27.0 (11.5) | 26.7 (11.3) | 27.0 (11.5) | 0.487 |
| F33 recurrent depressive disorder diagnosis (%) | 28.5 | 23.2 | 30.0 | <0.001 |
| HoNOS65+ mental and physical health problems (%)†‡ | | | | |
| Agitated behaviour | 15.9 | 16.2 | 15.8 | 0.775 |
| Non-accidental self-injury | 12.4 | 7.9 | 13.8 | <0.001 |
| Drug/alcohol problems | 5.6 | 4.4 | 6.0 | 0.100 |
| Hallucinations or delusions | 11.3 | 10.2 | 11.7 | 0.232 |
| Depressed mood | 67.4 | 61.8 | 69.1 | <0.001 |
| Physical illness or disability | 60.8 | 61.1 | 60.8 | 0.846 |
| Cognitive problems | 19.1 | 35.7 | 14.1 | <0.001 |
| HoNOS65+ functional problems (%)†‡ | | | | |
| Activities of daily living | 44.1 | 47.2 | 43.1 | 0.042 |
| Living conditions | 13.1 | 11.7 | 13.6 | 0.168 |
| Occupational and recreational activities | 36.0 | 35.0 | 36.3 | 0.526 |
| Social relationships | 25.2 | 21.9 | 26.2 | 0.015 |
| Psychotropic use: in 2 years before depression diagnosis | | | | |
| Antidepressant (%) | 56.1 | 55.6 | 56.3 | 0.720 |
| Psychotropic use: after depression diagnosis | | | | |
| Antidepressant (%) | 86.5 | 91.2 | 85.1 | <0.001 |
| SSRI (%) | 55.2 | 61.3 | 53.4 | <0.001 |
| Antipsychotic (%) | 34.4 | 38.5 | 33.2 | 0.006 |
| Cognitive behavioural therapy | | | | |
| Received CBT postdepression diagnosis (%) | 26.8 | 22.2 | 28.1 | 0.001 |

*$\chi^2$ test, analysis of variance or independent samples t-test.
†At or closest to the time of depression diagnosis.
‡Health of the Nation Outcome Scale 65+ (HoNOS65+) subscale scores 0–4 (0=least severe, 4=most severe status). Values represent frequencies of patients scored as experiencing problems (score 2–4) in that domain.
CBT, cognitive behavioural therapy; HoNOS65+, Health of the Nation Outcome Scales for older people; SSRI, selective serotonin reuptake inhibitor.

of dementia. We found that a diagnosis of recurrent depressive disorder was protective in relation to overall dementia risk, and in those who went on to develop dementia, recurrent depression was associated with a slower progression to a diagnosis which translated to over a year later than the average time. We postulate that the 187 patients with recurrent depression who developed dementia (23.2% of the total 806 patients who developed dementia) were more likely to be suffering initially from mood disorder distinct from incipient dementia, and so development of dementia overall came later. The remaining patients who were categorised as having new-onset depression may have been following a trajectory of newly emergent, steadily increasing depressive

**Table 2** Results from a series of multivariate Cox proportional hazards models analysing associations between predictor variables and risk of dementia diagnosis (adjusted HR (95% CI)) in 3659 patients with late-life depression, of whom 806 were subsequently diagnosed with dementia during study follow-up

| Predictor variables | Model 1<br>Adjusted for age, gender, deprivation | Model 2<br>Model 1+depressed mood†<br>+cognitive problems† | Model 3<br>Adjusted for all predictor variables |
|---|---|---|---|
| Demographics‡ | | | |
| Centred age (per year difference from mean) | 1.07 (1.06–1.08)** | 1.05 (1.04–1.06)** | 1.04 (1.03–1.06)** |
| *Centred age\*time§* | | *1.01 (1.00–1.01)\** | *1.01 (1.00–1.01)\** |
| Female gender | 0.95 (0.82–1.11) | 0.97 (0.83–1.12) | 0.99 (0.84–1.16) |
| Non-white ethnicity | 1.36 (1.13–1.62)** | 1.22 (1.02–1.46)* | 1.14 (0.95–1.37) |
| Married or cohabiting status | 1.07 (0.91–1.25) | 1.10 (0.94–1.29) | 1.09 (0.92–1.28) |
| Deprivation (per 10 unit increase in IMD score) | 1.01 (0.95–1.08) | 0.99 (0.94–1.06) | 0.99 (0.93–1.06) |
| Recurrent depressive disorder diagnosis or previous diagnosis of depression | 0.47 (0.36–0.62)** | 0.65 (0.55–0.77)** | 0.65 (0.55–0.77)** |
| *Recurrent depression\*time§* | *1.09 (1.02–1.16)\** | | |
| HoNOS65+ mental and physical health problems†‡ | | | |
| Agitated behaviour | 1.09 (0.90–1.33) | 0.87 (0.71–1.06) | 0.95 (0.77–1.17) |
| Non-accidental self-injury | 0.69 (0.53–0.88)* | 0.68 (0.52–0.88)* | 0.76 (0.59–0.99)* |
| Drug/alcohol problems | 1.03 (0.72–1.48) | 0.97 (0.68–1.40) | 1.09 (0.77–1.55) |
| Hallucinations and delusions | 0.88 (0.70–1.10) | 0.57 (0.39–0.83)* | 0.57 (0.38–0.84)* |
| *Hallucinations and delusions\*time§* | | *1.08 (0.99–1.18)* | *1.10 (1.00–1.21)\** |
| Depressed mood | 1.11 (0.96–1.28) | 1.02 (0.88–1.18) | 1.05 (0.90–1.24) |
| Physical illness or disability | 1.33 (1.15–1.54)** | 1.18 (1.02–1.38)* | 0.97 (0.76–1.23) |
| *Physical illness or disability\*time§* | | | *1.05 (0.99–1.12)* |
| Cognitive problems | 4.89 (3.84–6.21)** | 4.96 (3.89–6.32)** | 4.72 (3.67–6.06)** |
| *Cognitive problems\*time§* | *0.84 (0.78–0.91)\*\** | *0.84 (0.78–0.91)\*\** | *0.84 (0.77–0.91)\*\** |
| HoNOS65+ functional problems†‡ | | | |
| Activities of daily living | 1.43 (1.24–1.65)** | 1.12 (0.96–1.31) | 1.06 (0.90–1.26) |
| Living conditions | 1.04 (0.83–1.31) | 0.88 (0.69–1.12) | 0.87 (0.68–1.12) |
| Occupational/recreational activities | 1.13 (0.97–1.32) | 1.02 (0.87–1.19) | 1.06 (0.89–1.26) |
| Social relationships | 1.10 (0.85–1.42) | 1.03 (0.79–1.33) | 1.13 (0.86–1.48) |
| *Social relationships\*time§* | *0.92 (0.85–1.42)\** | *0.93 (0.87–1.00)\** | *0.92 (0.85–0.99)\** |
| Psychotropic use: in 2 years before depression diagnosis | | | |
| Antidepressant | 0.87 (0.76–1.00)* | 0.90 (0.78–1.04) | 0.93 (0.80–1.08) |
| Psychotropic use: after depression diagnosis | | | |
| Any antidepressant | 1.21 (0.95–1.55) | 1.27 (0.99–1.61) | 1.32 (1.01–1.74)* |
| SSRI | 1.12 (0.97–1.29) | 1.11 (0.96–1.28) | 1.07 (0.91–1.25) |
| Antipsychotic | 0.88 (0.76–1.02) | 0.85 (0.73–0.99)* | 0.98 (0.84–1.14) |
| Cognitive behavioural therapy | | | |
| Received CBT postdepression diagnosis | 0.62 (0.52–0.73)** | 0.69 (0.58–0.82)** | 0.73 (0.61–0.87)** |

*p<0.05; **p<0.002 (Bonferroni corrected).

†Health of the Nation Outcome Scale 65+ (HoNOS65+) subscale scores 0–4 (0=least severe, 4=most severe status). Values represent frequencies of patients scored as experiencing problems (score 2–4) in that domain.

‡At or closest to the time of depression diagnosis.

§Time-variable interactions describing how the hazard in the variable reported above changes per year.

CBT, cognitive behavioural therapy; HoNOS65+, Health of the Nation Outcome Scales for older people; SSRI, selective serotonin reuptake inhibitor.

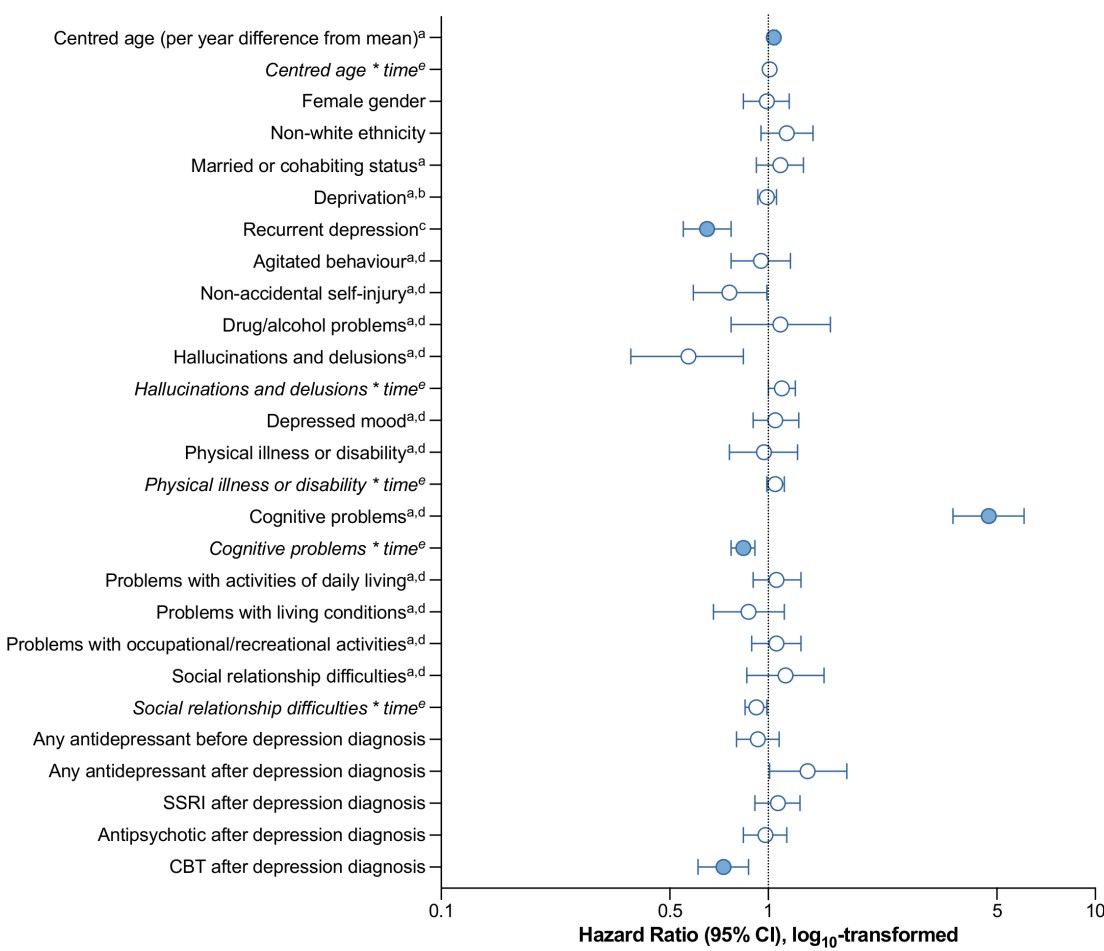

**Figure 2** Plot of HRs for risk of dementia diagnosis in a multivariate Cox regression model adjusted for all covariates in 3659 patients with late-life depression, of whom 806 developed dementia. Coloured points indicate **p<0.002 (Bonferroni corrected). [a]At or closest to the time of depression diagnosis. [b]Per 10 unit increase in IMD score. [c]Recurrent depressive disorder diagnosis or previous diagnosis of depression. [d]Health of the Nation Outcome Scale 65+ (HoNOS65+) subscale scores 0–4 (0=least severe, 4=most severe status). Values represent frequencies of patients scored as experiencing problems (score 2–4) in that domain. [e]Time-variable interactions describing how the hazard in the variable reported above changes per year. CBT, cognitive behavioural therapy; SSRI, selective serotonin reuptake inhibitor.

symptoms which have previously been shown to characterise prodromal dementia[7–9] and reached a threshold of severity, at which point they reached secondary mental health services and the index diagnosis of depression was made. Methodological differences complicate comparisons between findings relating to the relationship between recurrent depression and dementia risk, as studies vary in their classifications of new-onset and recurrent depression, the time frame, and whether it was self-reported or clinically diagnosed. In this study, the severity and persistence of clinical symptoms across the cohort warranted a referral to secondary mental health services. The results are consistent with the findings from another large, community-based cohort study, in showing that a recent episode of depression in later life is a strong predictor of progression to dementia.[29]

It is not surprising that more severe cognitive impairment in patients with late-life depression was associated with a higher dementia risk and predicted that a diagnosis of dementia would be made sooner. Past studies have shown that recently active depression in combination with mild cognitive impairment (MCI) confers particularly high risk for AD; 40% progressed to AD within an average follow-up time of just over 2 years.[29] Similarly, in a cohort of older adults with high depressive symptoms in German primary care services, coexistence of marked cognitive deficits was a distinguishing prognostic marker of subsequent dementia within the 6-year follow-up period.[30] Against this background, the authors discouraged the attribution of cognitive impairment in late-life depression to 'depressive pseudodementia' as it can be a strong indication of incipient dementia that warrants further investigation; a view supported by our study. A case–control sample from the PAQUID cohort demonstrated a similar emergence of depressive symptoms and cognitive and functional decline in the years preceding the dementia phase of AD.[31] In that cohort, progression of cognitive decline was accompanied by depressive symptoms, and as a consequence of continued worsening, independence in ADLs then began to deteriorate.

Regarding the potential effects of antidepressant medication, although not significant after accounting

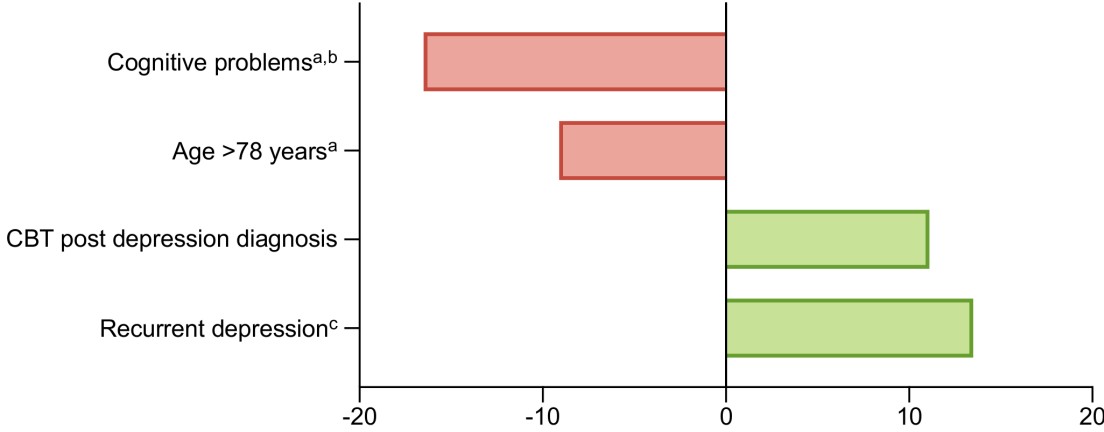

**Figure 3** Impact of significant risk factors on time to dementia diagnosis in 806 patients with late-life depression who were subsequently diagnosed with dementia during follow-up. [a]At or closest to the time of depression diagnosis. [b]Health of the Nation Outcome Scale 65+ (HoNOS65+) subscale scores 0–4 (0=least severe, 4=most severe status). Values represent frequencies of patients scored as experiencing problems (score 2–4) in that domain. [c]Recurrent depressive disorder diagnosis or previous diagnosis of depression. CBT, cognitive behavioural therapy.

for multiple comparisons, those prescribed 'any antidepressant' appeared to be at a higher risk of developing dementia in our cohort. This is in line with the growing evidence for this association from observational studies,[32] and is consistent with a large, matched, prospective cohort study that found that antidepressant monotherapy was associated with an over twofold increased dementia risk in those aged over 60 after adjusting for multiple confounders, including depression.[33] Poor antidepressant response has been reported to be indicative of an especially increased risk of dementia in patients with cognitive impairment,[34] and it is likely that a number of patients in our study were referred to secondary care following treatment failure because over half had been prescribed antidepressants prior to reaching the service. However, this is challenging to disentangle if the efficacy of antidepressants might generally be reduced in patients aged over 65, as suggested by a number of meta-analyses.[35 36] As over 85% of our cohort were prescribed antidepressant drugs following their depression diagnosis, the analysis lacked the comparison group required to elucidate whether this

indeed had an influence on dementia risk. Also, as only a crude measure of depression severity was adjusted for, it is possible that the few patients who were not prescribed an antidepressant might have suffered from milder forms of depression, meaning that the association with dementia risk reflects differences in depression symptomology rather than antidepressant treatment. Potentially of note is that the 'any antidepressant' group yielded an increased dementia risk, but the SSRI group did not. This could be due to a lack of statistical power as the number of patients prescribed specifically SSRIs was lower, or it could indicate that the inclusion of antidepressants with a higher anticholinergic burden in this group (eg, tricyclic antidepressants) might have contributed to the increased hazard of developing dementia.[37] Robust analyses of the efficacy of specific pharmacotherapies are often outside the scope of observational cohort studies, and from our study we cannot infer any neuroprotective effects of antidepressants. Controlled trials are yet to identify any candidate antidepressant treatments that can improve cognitive outcomes in older adults with depression,[38 39]

**Table 3** Mean time from depression diagnosis to dementia diagnosis per score increase on the HoNOS65+ cognitive problems subscale recorded at the time of depression diagnosis, in 806 patients who were subsequently diagnosed with dementia

| HoNOS65+ cognitive problems scores | Mean time from depression diagnosis to dementia diagnosis in years (95% CI) |
|---|---|
| 0=no problem | 3.77 (3.44–4.11) |
| 1=minor problem requiring no action | 2.94 (2.67–3.20) |
| 2=mild problem but definitely present | 2.05 (1.75–2.35) |
| 3=moderately severe problem | 1.81 (1.39–2.22) |
| 4=severe to very severe problem | 1.36 (0.25–2.46) |

HoNOS65+, Health of the Nation Outcome Scales for older people.

except for some small short-term improvements in executive functioning.[40 41]

After adjusting for multiple confounders, undergoing CBT was associated with an almost 30% reduced risk of later being diagnosed with dementia. One likely explanation is that the patients who were offered CBT systematically differed from those not offered the intervention beyond what was possible to control for through adjusting for confounders, particularly in relation to their severity of depressive and cognitive symptoms. On the other hand, cognitive behavioural techniques are known to reduce some of the typical risk factors for dementia such as lack of cognitive, physical or social activity, a restricted diet or sleep disorders.[42 43] Further, CBT for insomnia has shown benefits to executive functioning in a small randomised study of people with MCI and insomnia.[44] Whether CBT needs to be adapted for late-life depression and could show benefits to cognitive decline requires further exploration.[45]

Of note, although not significant after adjustment for multiple comparisons, the presence of non-accidental self-injury and psychotic symptoms appeared to be associated with a lower dementia risk. These results are consistent with a previous study of late-life depression that used this data source, in which certain depressive symptoms, that is, guilt feelings, impaired concentration, disturbed sleep and delusions, were associated with a lower mortality risk in people with late-life depression.[46] The study also showed that severity of problems related to depressed mood on the HoNOS65+ subscale predicted mortality only when patients with new-onset depression were considered.[46] This could be related to the sample being drawn from a secondary care provider for which a certain severity threshold needs to be met, and might explain why depression severity according to the HoNOS65+ subscale was not associated with an increased risk of dementia in the present study.

### Strengths and limitations

Strengths of this study include the large, population-based sample in routine clinical care. The main limitations relate to the generalisability of the findings drawn from this selective cohort. To be captured in the study patients needed a secondary care diagnosis of depression, meaning these findings might not generalise to patients with milder depressive symptoms treated in primary care or those not readily seeking a depression diagnosis.[47] Crucially, the findings from examining the duration between depression and dementia diagnosis relate to the time taken to receive a diagnosis of dementia in SLaM services, rather than the time taken to develop the diagnosable symptoms of dementia. This duration is influenced by potential barriers to timely diagnosis; service, provider, patient and caregiver factors contribute to delayed dementia diagnosis.[48] A review of the literature on missed and delayed diagnosis of dementia found symptom severity and degree of impairment are important predictors of diagnostic sensitivity.[48] Additionally, late-life depression confers an increased mortality risk, both in relation to suicide and when suicidality is accounted for,[49] and the subcohort of patients who received both depression and dementia diagnoses in SLaM was selective to the 806 patients who survived long enough to receive a dementia diagnosis.

Of the 806 patients diagnosed with dementia, we found that 25% received an initial diagnosis of unspecified dementia and only 44% received a diagnosis of AD. This is lower than the expected incidence of AD in two-thirds of dementia cases[50] and likely to reflect the naturalistic nature of the data, whereby the specific subtype is often recorded as unspecified until further investigations are conducted or more specific symptoms emerge. Subsequently, the recording of specific subtypes increases if the whole record is considered.[51 52]

Medication prescription was ascertained either within the 2 years before or after the index date of first depression diagnosis through natural language processing, but the time-dependent nature of prescribing could not be accounted for. Further, the output of natural language processing depends on the accuracy and quality of data entry, which varies by individual clinician and is compromised through the use of jargon, idiosyncratic abbreviations or misspellings.[17] Although it has been shown that precision and recall are relatively high for the medication natural language processing application,[17] there remains a risk underestimating or overestimating the true prescribing prevalence, especially as adherence to medication cannot be established.[53] Lastly, although we incorporated a wide range of potential confounders in our analysis, causal inferences cannot readily be drawn in an observational study of this nature.

### Conclusions

The aim of our study was to highlight clinical markers of increased risk and accelerated progression to dementia in late-life depression. A knowledge of the indicators of the preclinical phase of dementia may be crucial to aid advancement in the identification, and consequently the management, of the disease at the earliest possible stage. As patients at the earliest stage of dementia are most likely to benefit from intervention to delay progression, future efforts to improve the detection of dementia should focus on recognising subtle, early manifestations of disease. By examining the temporal relationships between characteristics within late-life depression and progression to dementia, in the context of South London secondary mental health service users, we have highlighted features that may indicate higher risk of and sooner progression to dementia. In older patients presenting with new-onset depression and/or more problematic cognitive symptoms, clinicians should suspect a diagnosis of dementia. It may be beneficial to closely monitor these patients for signs of cognitive deterioration suggestive of dementia to facilitate the earliest possible detection and diagnosis. These patients might also form an interesting group for

which novel pharmacological or psychosocial interventions could be developed.

**Contributors** GeP, CM, RS and DA conceived the study. GeP, CM and GaP performed the analysis and interpreted the data. GeP and CM wrote the manuscript. NK, MS, DA and RS provided expertise and feedback. All authors critically revised the manuscript and approved the final version of the study.

**Funding** This study was funded by Wellcome Trust.

**Competing interests** RS has received research funding from Roche, Pfizer, Janssen, Lundbeck and InSilico-Bioscience. DA has received research support and/or honoraria from AstraZeneca, H. Lundbeck, Novartis Pharmaceuticals and GE Health, and serves as paid consultant for H. Lundbeck and Axovant.

**Patient consent for publication** Not required.

**Provenance and peer review** Not commissioned; externally peer reviewed.

**Data availability statement** Deidentified patient data were obtained via the Clinical Record Interactive Search system (CRIS) at the National Institute for Health Research (NIHR) Biomedical Research Centre (BRC) at South London and Maudsley NHS Foundation Trust and the Institute of Psychiatry, Psychology & Neuroscience at King's College London. The data accessed by CRIS remain within an NHS firewall and governance is provided by a patient-led oversight committee. There are no additional unpublished data available.

**ORCID iDs**
Georgia Peakman http://orcid.org/0000-0002-3319-138X
Gayan Perera http://orcid.org/0000-0002-3414-303X

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
