## [Reviewer comments · BMJ Open]

ARTICLE DETAILS

TITLE (PROVISIONAL)	Clinical factors associated with progression to dementia in people with late-life depression: a cohort study of patients in secondary care
AUTHORS	Peakman, Georgia; Karunatilake, Nishshanka; Seynaeve, Mathieu; Perera, Gayan; Aarsland, Dag; Stewart, Robert; Mueller, Christoph

VERSION 1 – REVIEW

REVIEWER	Peter Watson University of Cambridge UK
REVIEW RETURNED	09-Dec-2019

GENERAL COMMENTS	Clinical factors associated with progression to dementia in people with late-life depression: a cohort study of patients in secondary care bmjopen-2019-035147 Page 10, lines 35-37. A rather naive comment, perhaps, but I assume since time from depression diagnosis to dementia is the outcome of interest that dementia always follows depression? Could you, for example, be diagnosed with dementia and then later have a diagnosis of depression? Page 11, lines 23-25. Am I correct here that the people in the analysis are solely the 806 with both a depression and dementia diagnosis as seems to be implied here? If so I can't see that since all the people in the analysis have an 'event' occurring namely dementia why you didn't just perform a linear regression with time from depression to dementia as the outcome variable since everyone in the analysis has experienced a diagnosis of dementia and thus has a time when dementia took place so you have no need for a Cox Model since you have no censored observations (ie there are no people in the analysis who did not have a diagnosis of dementia in the time frame of the study). In fact a better approach would be to additionally take advantage of the extra information in censored observations (people with depression but who have not as yet being diagnosed with dementia but are assumed to definitely get dementia at some point) and use the Cox Model on all 5910 patients diagnosed with depression including the 5104 (Page 11, lines 14-19) who had received a diagnosis of depression but had not as yet received a diagnosis of dementia. You would put time from depression diagnosis as the outcome with a binary censored indicator (either no dementia diagnosed by that time or dementia was diagnosed at that time for each patient). I would add the sample size (N=806 or
---

	N=5910 or other) in Table 2 (Page 15) as is done in Table 1 (Page 12) to clarify who are the people in the Cox Model analysis. Page 15. The results for the three models look very similar. I would therefore just fit Model 3 with contains all the covariates and which appears to address the associations of interest as reported in the abstract on page 3. I assume you have tested the proportional hazards assumption underlying the use of covariates with the Cox Proportional Hazards Model? Namely that the hazard rate over time is the same for different covariate subgroups e.g. males and females. I assume that the covariate values do not change over time e.g. married or cohabiting status has not changed over the time period. If there are covariates changing over the time period would you consider fitting them as time varying covariates? I also wondered, on a similar theme if you could have alternatively modelled the depression covariate as a time-varying covariate since people who have been diagnosed as depressed would have come out of depression at a later date hence their depression status would have changed over time.
--	--

REVIEWER	Lynette Chenoweth University of New South Wales, Australia
REVIEW RETURNED	20-Dec-2019

GENERAL COMMENTS	The study objectives and procedures are clearly described, the statistical methods are suitable and the results answer the research questions. The manuscript is well written and logically structured. The Discussion of study findings is particularly thoughtful and insightful. I cannot recommend any improvements that are required.
--

REVIEWER	Prof Stephen Z Levine (PhD) University of Haifa
REVIEW RETURNED	23-Jan-2020

GENERAL COMMENTS	This manuscript examines factors associated with dementia among persons with depression. Despite my remarks, I think this is an interesting contribution to this domain of research.  1. The abstract states– “Depression in later life is increasingly being recognized as a prodrome for the development of dementia” – this should read “There is evidence that depression is a prodrome factor for the risk of dementia” – the jury is not out and it may be a risk factor. 2. The manuscript might benefit by mentioning the projected increase in depression.¹ 3. In the context of depression studies I do not think 2.8y is “relatively long” compared with high-quality epidemiology studies (not clinical trials). 4. In today’s age of big data I would not say this is the accurate– “unique access to anonymized clinical data” remove “unique”. 5. The authors should re-evaluate the evidence regarding antidepressants. They cite the Lancet 2017 paper, yet ignored the WHO guidelines and large observational studies. First, recent large observational studies find increased risk (uncited so cite2).
---

	This finding is consistent with meta-analysis³. Second the WHO guidelines state there is no evidence for neuroprotective effects of antidepressants. Third the efficacy of antidepressants in old-age is limited.^{4,5} Therefore how can antidepressants be modifiable through antidepressants. CBT we hope. 6. Clarify this is actually incident dementia. 7. How were the competing risks of dementia and mortality accounted for. 8. How were violations of the assumption of proportional hazards accounted for? Maybe a Zou Poisson model would be superior here? 9. The way the medications are analyzed does not address their time-depend nature. 10. Age – often studies add a linear and quadratic terms to account for the change in risk with age. 11. Are these prescribed / prescribed & purchased / reported antidepressant use? Each method with their own restrictions. 12. Use FDR P-values owing to multiple tests. 13. Models of increasing complexity – Good approach - please include the BICs and other information fit indices & just report the most parsimonious model. 14. Please expand on this - 353 (43.8%) developed Alzheimer’s disease. If we believe the literature AD accounts for far more cases of dementia than this figure. 15. Confounders are inadequately accounted for and the authors correctly acknowledge this shortcoming. 16. Considering the suicide rate of old people with depression, is selection mortality possible? Thank you for allowing me to review your work. References 1. Heo M, Murphy CF, Fontaine KR, et al. Population Projection of Us Adults with Lifetime Experience of Depressive Disorder by Age and Sex from Year 2005 to 2050. Int J Geriatr Psychiatry. 2008;23(12):1266-1270. 2. Kodesh A, Sandin S, Reichenberg A, et al. Exposure to Antidepressant Medication and the Risk of Incident Dementia. Am J Geriatr Psychiatry. 2019;27(11):1177-1188. 3. Wang Y-C, Tai P-A, Poly TN, et al. Increased Risk of Dementia in Patients with Antidepressants: A Meta-Analysis of Observational Studies. Behavioural neurology. 2018;2018. 4. Tham A, Jonsson U, Andersson G, et al. Efficacy and Tolerability of Antidepressants in People Aged 65 Years or Older with Major Depressive Disorder - a Systematic Review and a Meta-Analysis. J Affect Disord. 2016;205:1-12. 5. Tedeschini E, Levkovitz Y, Iovieno N, et al. Efficacy of Antidepressants for Late-Life Depression: A Meta-Analysis and Meta-Regression of Placebo-Controlled Randomized Trials. J Clin Psychiatry. 2011;72(12):1660-1668.
--	--

VERSION 1 – AUTHOR RESPONSE

Reviewer 1:	Response	Changes in Manuscript
1. Page 10, lines 35-37. A rather naive comment,	This is a very interesting observation. While depression as a 'functional'	

perhaps, but I assume since time from depression diagnosis to dementia is the outcome of interest that dementia always follows depression? Could you, for example, be diagnosed with dementia and then later have a diagnosis of depression?	mental illness is considered treatable until full recovery, this is not possible for the neurodegenerative condition dementia. Hence for dementia, it becomes even more important to address potential risk factors, such as depression. People with dementia can develop depressive symptoms, but these are considered to be part of the dementia syndrome and neurodegenerative process, rather than a separate entity. The underlying pathophysiology of depression in dementia remains somewhat unclear; with e.g. evidence for a lack of antidepressant efficacy versus placebo in dementia, and the search for potential treatments continues.	
2. Page 11, lines 23-25. Am I correct here that the people in the analysis are solely the 806 with both a depression and dementia diagnosis as seems to be implied here? If so I can't see that since all the people in the analysis have an 'event' occurring namely dementia why you didn't just perform a linear regression with time from depression to dementia as the outcome variable since everyone in the analysis has experienced a diagnosis of dementia and thus has a time when dementia took place so you have no need for a Cox Model since you have no censored observations (ie there are no people in the analysis who did not have a diagnosis of dementia in the time frame of the study). In fact a better approach would be to additionally take advantage of the extra information in censored observations (people with depression but who have not as yet being diagnosed with dementia but are assumed to definitely get dementia at some point)	Many thanks for those interesting observations and suggestions, which have led to us reconsider our approach. As you highlighted, we didn't want to lose all the observation from the censored observation. As censoring point, we chose the last clinical contact with mental health services, as this is a point when we can be reasonably certain that the person has not (yet) developed dementia. In line with the dementia ascertainment, we only included patients who were followed-up for at least 3 months, which changed the numbers slightly. Missingness was higher in those not developing dementia, leading to the necessity to impute missing values. This affects all chapters of the manuscript, and we have highlighted these in yellow. Table 1 now also compares those who developed dementia and those censored, making Supplementary Table 1 no longer necessary.	All chapters

and use the Cox Model on all 5910 patients diagnosed with depression including the 5104 (Page 11, lines 14-19) who had received a diagnosis of depression but had not as yet received a diagnosis of dementia. You would put time from depression diagnosis as the outcome with a binary censored indicator (either no dementia diagnosed by that time or dementia was diagnosed at that time for each patient). I would add the sample size (N=806 or N=5910 or other) in Table 2 (Page 15) as is done in Table 1 (Page 12) to clarify who are the people in the Cox Model analysis.		
3. Page 15. The results for the three models look very similar. I would therefore just fit Model 3 with contains all the covariates and which appears to address the associations of interest as reported in the abstract on page 3.	Our three models remain similar, but we still think there is some value for the reader to see all three models; especially as following the suggestions from Reviewer 2, we have clarified whether a significance value would hold following correction for multiple comparisons.	Table 2
4. I assume you have tested the proportional hazards assumption underlying the use of covariates with the Cox Proportional Hazards Model? Namely that the hazard rate over time is the same for different covariate subgroups e.g. males and females.	We checked proportional hazard assumptions and added variables as time-varying confounders should those be violated.	Methods – Statistical analysis
5. I assume that the covariate values do not change over time e.g. married or cohabiting status has not changed over the time period. If there are covariates changing over the time period would you consider fitting them as time varying covariates? I also wondered, on a similar theme if you could have alternatively modelled the depression covariate as a time-varying covariate since people who have been diagnosed as depressed would have come out of	Co-variates are only ascertained at the time of depression diagnosis, should their hazard vary over time as determined above, we included them as time-varying covariates.	Methods Statistical analysis

depression at a later date hence their depression status would have changed over time.		
Reviewer 2:		
The study objectives and procedures are clearly described, the statistical methods are suitable, and the results answer the research questions. The manuscript is well written and logically structured. The Discussion of study findings is particularly thoughtful and insightful. I cannot recommend any improvements that are required.	Many thanks for the positive feedback and very encouraging comments.	
Reviewer 3:		
1. The abstract states– “Depression in later life is increasingly being recognized as a prodrome for the development of dementia” – this should read “There is evidence that depression is a prodrome factor for the risk of dementia” – the jury is not out and it may be a risk factor.	Many thanks, we updated the abstract to reflect this uncertainty.	Abstract
2. The manuscript might benefit by mentioning the projected increase in depression. ¹	Many thanks for highlighting this, this certainly adds to the importance of our study and we have added this to the introduction.	Introduction
3. In the context of depression studies I do not think 2.8y is “relatively long” compared with high-quality epidemiology studies (not clinical trials).	Many thanks, we have removed this point from the article summary.	Article summary
4. In today’s age of big data I would not say this is the accurate– “unique access to anonymized clinical data” remove “unique”.	This is a valid point; many thanks; we have removed the term ‘unique’	Article summary

5. The authors should re-evaluate the evidence regarding antidepressants. They cite the Lancet 2017 paper, yet ignored the WHO guidelines and large observational studies. First, recent large observational studies find increased risk (uncited so cite²). This finding is consistent with meta-analysis³. Second the WHO guidelines state there is no evidence for neuroprotective effects of antidepressants. Third the efficacy of antidepressants in old-age is limited.^{4,5} Therefore how can antidepressants be modifiable through antidepressants. CBT we hope.	Many thanks for pointing us to these important resources, which has facilitated a rewrite of the respective section of the discussion.	Discussion
6. Clarify this is actually incident dementia.	We added this to the abstract and the methods.	Abstract, Methods
7. How were the competing risks of dementia and mortality accounted for.	We used Cox regression models to account for censoring due mortality or end of follow-up time. We changed our approach in response to suggestion 2 by Reviewer 2, and clarified in the methods: Date of first depression diagnosis after the age of 65 years served as the index date. Patients were followed up until the date of the first dementia diagnosis (incident dementia) given in the health record or until the last face-to-face contact with a professional from secondary mental health services (clinical event or appointment). Further, follow-up also ended at death or a censoring point on 30/06/2017. Patients were excluded if they had a recorded dementia diagnosis before or within 3 months after index date or if they had less than 3 months of contact with SLAM services after a depression diagnosis.	Methods
8. How were violations of the assumption of proportional hazards accounted for? Maybe a Zou Poisson model would be superior here?	We prefer the Cox proportional hazard model, as not all covariates are binary, and the Cox model has less assumptions. We tested for violations of the proportional hazard	Methods

	assumptions and added variables as time-varying if they were violated.	
11. Are these prescribed / prescribed & purchased / reported antidepressant use? Each method with their own restrictions.	Medication prescription is ascertained via natural language processing from free text. Together with being unable to account for the time-dependent nature of prescribing, we acknowledged the drawbacks of this approach as follows in the limitations: Medication prescription was ascertained either within the 2 years before or after the index date of first depression diagnosis through natural language processing, but the time-dependent nature of prescribing could not be accounted for. Further, the output of natural language processing depends on the accuracy and quality of data entry, which varies by individual clinician and is compromised through the use of jargon, idiosyncratic abbreviations or misspellings¹⁷. Although it has been shown that precision and recall are relatively high for the medication natural language processing application¹⁷, there remains a risk under- or overestimating the true prescribing prevalence, especially as adherence to medication cannot be established⁵⁴.	Discussion - Limitations
9. The way the medications are analyzed does not address their time-dependent nature.		
10. Age – often studies add a linear and quadratic terms to account for the change in risk with age.	Many thanks, this is a great suggestion. We conducted a sensitivity analysis and added the following to the Results: Adding age as a squared term to the fully adjusted model (Model 3) didn't affect the predictors yielding significance, with the exception of non-accidental self-injury (attenuated to a trend; $p=.076$).	Results
12. Use FDR P-values owing to multiple tests.	Also, many thanks for making this important point. For Table 2 we now report $p<.05$ and a Bonferroni corrected p -value of $p<.002$.	Methods, Results, Table 2
13. Models of increasing complexity – Good approach -	As we are assessing various predictors, we'd like to report all	

please include the BICs and other information fit indices & just report the most parsimonious model.	models. We have however used the robust estimate of variance to account for the same subjects appear repeatedly in the risk pools.	
14. Please expand on this - 353 (43.8%) developed Alzheimer's disease. If we believe the literature AD accounts for far more cases of dementia than this figure.	This is because we only look at first recorded dementia subtype; to explain this we added the following to the discussion: Of the 806 patients diagnosed with dementia, we found that 25% received an initial diagnosis of unspecified dementia and only 44% a diagnosis of Alzheimer's disease. This is lower than the expected incidence of Alzheimer's disease in two-thirds of dementia cases⁵¹ and likely to reflect the naturalistic nature of the data, whereby the specific subtype is often recorded as unspecified until further investigations are conducted or more specific symptoms emerge. Subsequently the recording of specific subtypes increases if the whole record is considered^{52 53}.	Discussion - limitations
15. Confounders are inadequately accounted for and the authors correctly acknowledge this shortcoming.	Many thanks, we have now made this clearer in the limitations, in addition to the Highlights section.	Article summary, Discussion - Limitations
16. Considering the suicide rate of old people with depression, is selection mortality possible?	That is an important point. In initial Cox analyses we consider mortality as censoring point, but not in the sub-cohort of 806 patients who developed dementia. We have acknowledged this in the limitations as follows: Additionally, late-life depression confers an increased mortality risk, both in relation to suicide and when suicidality is accounted for⁵⁰, and the sub-cohort of patients who received both depression and dementia diagnoses in SLAM was selective to the 806 patients who survived long enough to receive a dementia diagnosis.	Discussion - Limitations
1. Heo M, Murphy CF,		

Fontaine KR, et al. Population Projection of Us Adults with Lifetime Experience of Depressive Disorder by Age and Sex from Year 2005 to 2050. Int J Geriatr Psychiatry. 2008;23(12):1266-1270. 2. Kodesh A, Sandin S, Reichenberg A, et al. Exposure to Antidepressant Medication and the Risk of Incident Dementia. Am J Geriatr Psychiatry. 2019;27(11):1177-1188. 3. Wang Y-C, Tai P-A, Poly TN, et al. Increased Risk of Dementia in Patients with Antidepressants: A Meta-Analysis of Observational Studies. Behavioural neurology. 2018;2018. 4. Tham A, Jonsson U, Andersson G, et al. Efficacy and Tolerability of Antidepressants in People Aged 65 Years or Older with Major Depressive Disorder - a Systematic Review and a Meta-Analysis. J Affect Disord. 2016;205:1-12. 5. Tedeschini E, Levkovitz Y, Iovieno N, et al. Efficacy of Antidepressants for Late-Life Depression: A Meta-Analysis and Meta-Regression of Placebo-Controlled Randomized Trials. J Clin Psychiatry. 2011;72(12):1660-1668.		
--	--	--

VERSION 2 – REVIEW

REVIEWER	Peter Watson University of Cambridge UK
REVIEW RETURNED	03-Mar-2020

GENERAL COMMENTS	Clinical factors associated with progression to dementia in people with late-life depression: a cohort study of patients in secondary care bmjopen-2019-035147.R1 This reads well. I just have a few remaining small points of clarification below. Page 8, lines 42-51. You could state here, if my understanding is correct, that the censored times correspond to all events bar the occurrence of the participant's first diagnosis of dementia since you are interested in times from depression to dementia so dementia is the event of interest. Page 10, lines 35-38. I don't see any further reference to the use of time varying covariates in the results but, if time varying covariates were used, did this complicate the results in that my understanding is that with time varying covariates the hazard rate varies with time to dementia depending upon values of the covariate which would mean you had to look at strata subgroups of the covariate separately to assess hazard rate? Page 10, lines 46-50. Does it, therefore, follow from the creation of 19 imputed data sets that the results in Tables 1 (descriptives), 2 (Cox model) and 3 (cognitive score mean times to dementia) are based upon pooled estimates averaged across the multiple imputations? If so, what procedure did you use in STATA to do the pooling across imputations? A brief mention of this in the paper would help understanding of what you have done. I have not come across multiple imputation using the Cox model before and am not familiar with the multiple imputation references given in the paper but I would imagine you can pool estimates and their covariance matrices across the imputations as you would do for any other regression model. Page 14, lines 11-12 and 27-38 and Page 15, Table 2 and line 47. Slight confusion here in that you mention that you are only considering p-values less than 0.002 as significant to account for multiple comparisons (Page 14, lines 11-12) yet you go on to flag two variables, having hallucinations and taking antidepressants which both have p-values above 0.002 as associated with lower dementia and then say (Page 20, lines 50-53) that using multiple comparison corrections are conservative. Page 15, Table 2 line 14. I wondered if you had centred age (ie subtracted its mean) to make its incremental effect as measured by the Cox Model more interpretable? If you do this then you are automatically comparing changes of one year increases in the hazard rate to the more intuitive average age rather than age zero (birth). Page 30, Figure 2. I don't see mention in the figure of any footnotes explaining to what the 'a' and 'd' superscripts alongside some of the variables mean.
---

REVIEWER	Stephen Levine University of Haifa
REVIEW RETURNED	26-Feb-2020

GENERAL COMMENTS	The authors were responsive to my remarks. I have no further remarks
---

	Thank you for allowing me to review this interesting work
--	---

VERSION 2 – AUTHOR RESPONSE

Reviewer 1	Response:	Changes in Manuscript
This reads well. I just have a few remaining small points of clarification below.	Thank you again for your thorough reading of our manuscript. We highly appreciate the time, effort, and attention to detail that you have provided.	
Page 8, lines 42-51. You could state here, if my understanding is correct, that the censored times correspond to all events bar the occurrence of the participant's first diagnosis of dementia since you are interested in times from depression to dementia so dementia is the event of interest.	Many thanks for the suggestion. Yes, that's correct, so we have added the suggested wording to the methods section to make this clearer.	Methods - Sample
Page 10, lines 35-38. I don't see any further reference to the use of time varying covariates in the results but, if time varying covariates were used, did this complicate the results in that my understanding is that with time varying covariates the hazard rate varies with time to dementia depending upon values of the covariate which would mean you had to look at strata subgroups of the covariate separately to assess hazard rate?	This is good point; for variables for which the proportionate hazard assumptions were violated, we included a variable*time interaction. We initially chose not to present this interaction as this might confuse some readers, but you have highlighted that it is more accurate to present the interactions in the tables and results. As it is difficult to check for time-varying confounding in imputed data, we checked this in the complete-case dataset - the findings are now reported in additional supplementary table.	Abstract Methods – Statistical analysis Results Table 2 Figure 2 Supplementary Table 1
Page 10, lines 46-50. Does it, therefore, follow from the creation of 19 imputed data sets that the results in Tables 1 (descriptives), 2 (Cox model) and 3 (cognitive score mean times to dementia) are based upon pooled estimates averaged across the multiple	Yes, that is correct. We have now clarified in the methods that we used the STATA mi package and combined co-efficients using Rubin's rules.	Methods – Statistical Analysis

imputations? If so, what procedure did you use in STATA to do the pooling across imputations? A brief mention of this in the paper would help understanding of what you have done. I have not come across multiple imputation using the Cox model before and am not familiar with the multiple imputation references given in the paper but I would imagine you can pool estimates and their covariance matrices across the imputations as you would do for any other regression model.		
Page 14, lines 11-12 and 27-38 and Page 15, Table 2 and line 47. Slight confusion here in that you mention that you are only considering p-values less than 0.002 as significant to account for multiple comparisons (Page 14, lines 11-12) yet you go on to flag two variables, having hallucinations and taking antidepressants which both have p-values above 0.002 as associated with lower dementia and then say (Page 20, lines 50-53) that using multiple comparison corrections are conservative.	Many thanks, this is an important observation. Although having hallucinations and taking antidepressants were not significant predictors after accounting for multiple comparisons, we felt that they were nonetheless worth some discussion. To avoid confusion, we have removed the sentence referring to FDR procedures as being conservative in survival analyses.	Discussion – final paragraph
Page 15, Table 2 line 14. I wondered if you had centred age (ie subtracted its mean) to make its incremental effect as measured by the Cox Model more interpretable? If you do this then you are automatically comparing changes of one year increases in the hazard rate to the more intuitive average age rather than age zero (birth).	We hadn't done this as yet, but it is a good idea. It is surprising that this is not done more often, and so we have added an explanation to the manuscript.	Methods – Covariates Results Table 2

Page 30, Figure 2. I don't see mention in the figure of any footnotes explaining to what the 'a' and 'd' superscripts alongside some of the variables mean.	Apologies, the footnotes/legends for the figures can be found at the end of the main text document and had been entered in the submission portal, but these are not displayed with the figures themselves at this stage. We have been advised by the editorial office that this is the correct approach for submission.	Figure 2 Figure 3
--	--	------------------------------